# Synthesis and Application of Domestic Glassy Carbon TiO₂ Nanocomposite for Electrocatalytic Triclosan Detection

Vesna Stanković [1] , Dragan Manojlović [2,3] , Goran M. Roglić [2], Dmitry S. Tolstoguzov [3], Dmitry A. Zherebtsov [3] , Daniel A. Uchaev [3], Viacheslav V. Avdin [3] and Dalibor M. Stanković [2,*]

1   Institute of Chemistry, Technology and Metallurgy-National Institute of the Republic of Serbia, University of Belgrade, Njegoševa 12, 11000 Belgrade, Serbia
2   Faculty of Chemistry, University of Belgrade, Studentski trg 12–16, 11000 Belgrade, Serbia
3   South Ural State University, Lenin Prospekt 76, 454080 Chelyabinsk, Russia
*   Correspondence: dalibors@chem.bg.ac.rs

**Abstract:** Nanoparticles of TiO₂ are suitable for many catalytic and photocatalytic applications due to their extraordinary properties such as superhydrophobicity, semiconductivity, electron-rich, and environmental compatibility. The main crystalline phases of TiO₂, anatase, and rutile possess different crystal structures, crystallinity, crystalline sizes, and specific surface areas, and these characteristics directly affect the catalytic performance of TiO₂. In the present study, domestic carbon material enhanced with TiO₂ nanoparticles was synthesized and used for the construction of a modified carbon paste electrode. The electrocatalytic activity of the modified electrodes was investigated depending on the TiO₂ crystalline phases in the electrode material. Furthermore, the obtained working electrode was utilized for triclosan detection. Under optimized experimental conditions, the developed electrode showed a submicromolar triclosan detection limit of 0.07 μM and a wide linear range of 0.1 to 15 μM. The relative standard deviations for repeatability and reproducibility were lower than 4.1%, and with satisfactory selectivity, the proposed system was successfully applied to triclosan monitoring in groundwater. All these results confirm that the sustainable production of new and domestically prepared materials is of great benefit in the field of electrocatalysis and that the morphology of such produced materials is strongly related to their catalytic properties.

**Keywords:** titanium dioxide; electrocatalysis; carbon paste electrode; triclosan; irgasan

## 1. Introduction

The unique physical and chemical properties of metals and metal oxide nanoparticles have enabled their use in the development of new electrochemical sensors. Transition metal oxide nanoparticles possess good electrical and photocatalytic properties because of their size, shape, stability, and larger surface area [1]. Among them, nanoparticles of TiO₂ have received much attention since they are suitable for many catalytic and photocatalytic applications due to their extraordinary properties such as superhydrophobicity, semiconductivity, electron-rich, and environmental compatibility [2–6].

TiO₂ nanoparticles are used as potential electrode modifiers, as they are low cost, reliable, highly conductive, and biocompatible. In the literature, a large number of papers can be found describing the use of TiO₂ as a modifier of working electrodes that were used for the detection of various biologically significant compounds [7–12]. Although the mentioned studies showed that TiO₂ could improve the capabilities of the electrode, they have a common drawback, which is that they do not take into account the influence of TiO₂ crystalline phases on electrocatalytic activity.

The main crystalline phases of TiO₂ are anatase and rutile, and they exhibit different specific physical properties, bandgaps, and electronic surface states. Rutile has a narrower bandgap (3.05 eV) than anatase (3.26 eV) [13–15]. The rutile phase is the most thermodynamically stable polymorph at any temperature. However, anatase is usually a primarily

formed phase, and its recrystallization to rutile does not occur below 600 °C [15]. Its crystal structure, crystallinity, crystalline size, and specific surface area directly affect the catalytic performance of $TiO_2$ [16]. The degree of crystallinity of $TiO_2$ is very important for its catalytic activity since the structural defects act as recombination centers [17]. Rutile has a lower surface area than anatase due to the larger size of the crystals, which influences its lower photoactivity. Many reports indicate that the catalytic activity of a mixture of both polymorphs is more active than that of anatase, while anatase is usually more active than rutile [17]. In such a heterojunction, anatase acts as an electron acceptor. It can be concluded that the catalytic properties of $TiO_2$ are conditioned by reactions on the surface of $TiO_2$ as well by the ratio of rutile and anatase phases in mixed-phase $TiO_2$ [18].

$TiO_2$ preparation methods can be divided into gas phases or solution methods. Flame pyrolysis is the most economically effective and is used for producing the well-known photocatalyst Degussa Evonic P25 [17]. Post-treatment of $TiO_2$ nanoparticles with benzene vapors at 300–500 °C was reported to produce a composite with 0.37–1.69 wt.% carbon [19]. For similar purposes, many carbon precursors such as n-hexane, poly(vinyl alcohol) or poly(ethylene terephthalate), citric acid, cellulose, glucose, sucrose, poly(divinylbenzene), and formaldehyde polymers may be used [19]. Solution methods provide a much more flexible and controllable way of obtaining $TiO_2$ as powders, monoliths, or films. Such a sol-gel method utilizes titanium alkoxides: $Ti(OPri)_4$, $Ti(OBu)_4$, or $Ti(OEt)_4$, which easily hydrolyze in the presence of water. It was reported that a $TiO_2$/C nanocomposite prepared from a mixture of $Ti(OBu)_4$ and glycerol was heated to 250 °C, but at this temperature, it is hard to call carbon a composite [20]. Preparing a $TiO_2$/C nanocomposite from a solution of $Ti(OBu)_4$ in furfural alcohol with a nonionogenic surfactant was recently described [21,22]. This method, in spite of others, can produce a clear solution of reagents at a wide variety of concentrations, approving 5–15 nm anatase particle formation and its uniform distribution in a carbon matrix. The method is suitable for bulk monolith or film production, which is important for the high electric conductivity of composite electrodes.

Triclosan (TCS), commercially named irgasan, is an antimicrobial and antifungal compound that is commonly added as an ingredient to many consumer sanitary products, such as antibacterial soaps and body washes, toothpaste, and some cosmetics. It also can be found in clothing, kitchenware, furniture, and toys [23]. Since many of the aforementioned products are widely used and applied to external surfaces of the human body, this chemical is typically released into domestic wastewater systems. Given its widespread usage, it is not surprising that the concentration of TCS in groundwater, as well as in soil, is continually rising due to its poor solubilization and effective accumulation, consequently posing a threat to human health and the environment. Although TCS is not considered as toxic as other organic pollutants, it might accumulate in the human body for periods, posing long-duration health risks [24]. Another major problem of TCS is that its degradation results in other harmful products such as methyl-triclosan, whose lipophilicity and resistance to biodegradation and photolysis make this metabolite even more dangerous to the environment than TCS [25]. Moreover, the photodegradation of TCS in aqueous solutions can produce hazardous chlorophenols and low chlorinated dioxins—substances much more toxic than the precursor [26]

There are numerous methods for TCS detection, such as gas chromatography [27,28] or high-pressure liquid chromatography [29–31]. These types of classic analytical methods demand expensive instruments and often require complex sample preparation, including extended extraction of the target analyte. On the other hand, electrochemical methods are characterized by their efficiency, speed, and simplicity of operation, as well as relatively cheap instrumentation [32–35]. Therefore, great attention has been devoted to the development of new electrochemical sensors for the detection of various biologically significant compounds, including TCS. To date, a certain number of different electrochemical sensors for the detection of TCS have been reported. These sensors are based on the modification of electrodes, such as GCE [36] or CPE [37], with different kinds of nanoparticles—carbon [36,38], metals [39], or metal oxides [37,38].

Currently, considerable attention is focused on optimizing the production of carbon material in order to achieve morphological changes that would lead to the creation of more porous surfaces and an improvement in the mechanical and electrical properties of electrode material. In this sense, we present the synthesis of domestic carbon material, enhanced with different amounts of $TiO_2$ nanoparticles, and utilized for the fabrication of a modified carbon paste electrode that is further used for TCS detection. Anatase is the only $TiO_2$ phase in one composite sample (IS29) and is a dominant phase in another sample (IS24). The composite material with a bulk glassy carbon matrix and anatase nanoparticles has not been explored before. The proposed method of composite production ensures excellent mechanical and galvanic contact between the conductive inert matrix and catalytic nanoparticles, both of which are crucial for the performance of composites in electrochemistry. With this in mind, the catalytic capabilities of the working electrodes were investigated depending on the main crystalline phase of $TiO_2$ in the electrode modification material.

## 2. Results and Discussion

### 2.1. Morphological Characterization of the Materials

$TiO_2$/C nanocomposite with 0.37 wt.% carbon had 2.5 times higher effectivity of phenol photocatalytic oxidation compared to pure $TiO_2$ [20]. Sample $TiO_2$/C with 1.2 mass% carbon exhibited an extension of the visible light photocatalytic activity wavelength over 700 nm compared with only 450 nm for P25 [22]. Transmission electron microscopy revealed uniformly distributed 5–15 nm $TiO_2$ particles in the carbon matrix (Figure 1).

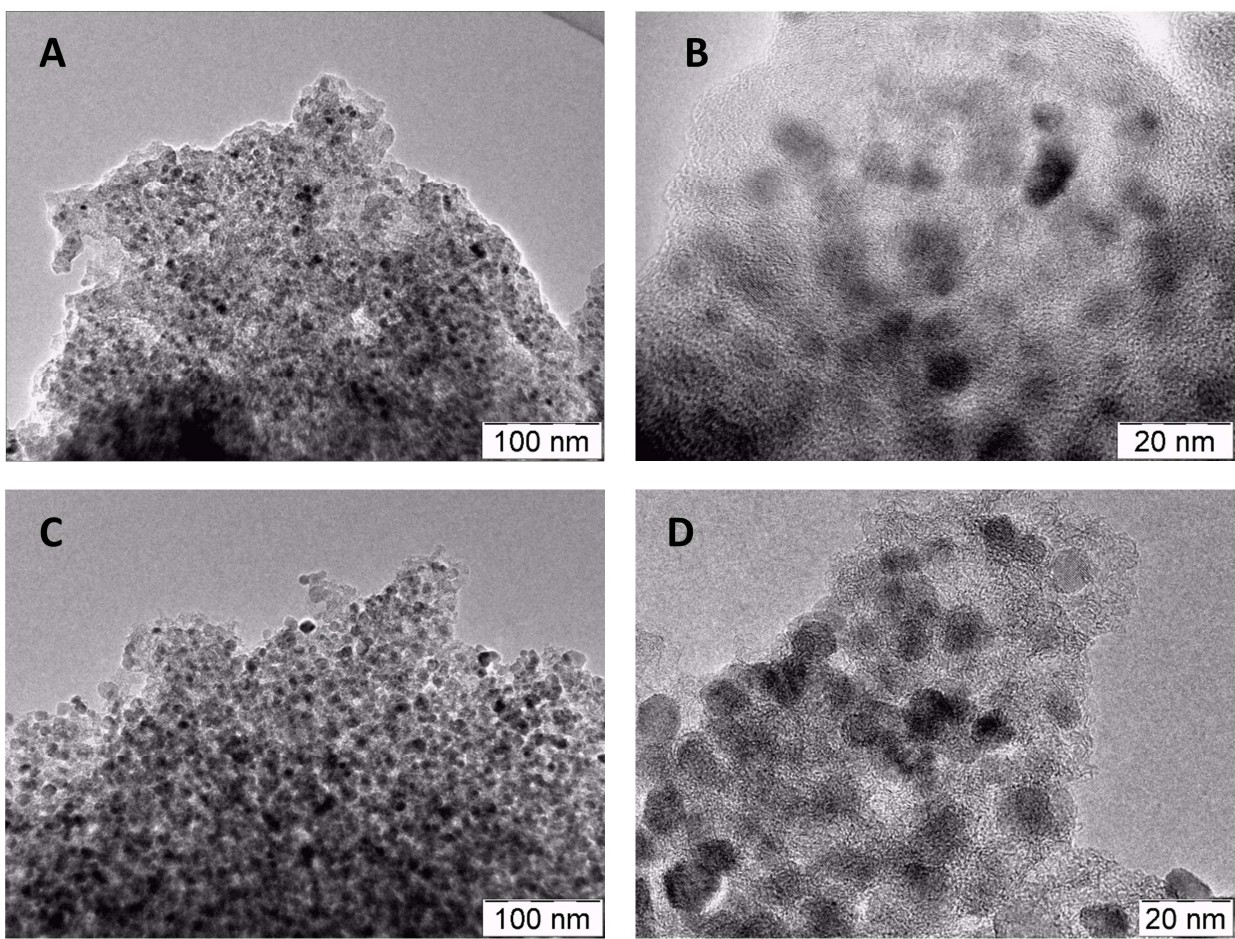

**Figure 1.** TEM image of IS24 (**A**,**B**) and IS29 (**C**,**D**).

Scanning electron microscopy reflected a uniform glassy carbon morphology of the composite fracture surface (Figure 2A,B). Powder X-ray diffraction confirmed the formation

of anatase and glassy carbon as the main phases (Figure 2C). Anatase average crystallite size calculated from the broadening of reflections on the diffractogram was 6.8–7.0 nm (8.7 nm by SAXS) for IS24 and 4.4–4.7 nm (6.1 nm by SAXS) for IS29. Besides anatase, sample IS24 contained about 6 wt.% rutile phases with a crystallite size of 20 nm. The micro- and mesoporosity of the material resulted in high adsorption of benzene vapors of 8.7 g/100 g for IS24 and 4.5 g/100 g for IS29.

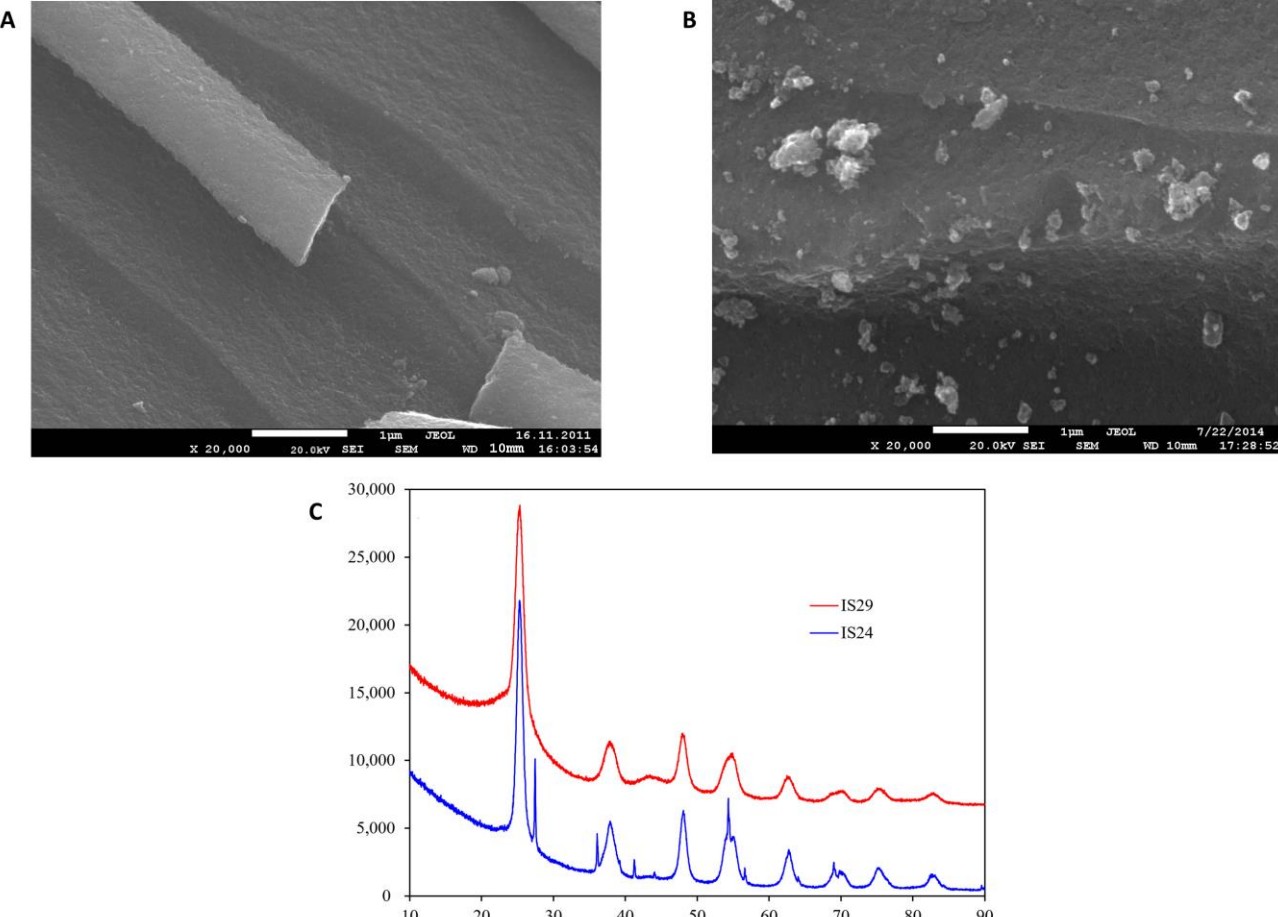

**Figure 2.** (**A**) SEM image of IS24; (**B**) SEM image of IS29; (**C**) powder X-ray diffraction of samples.

It should be noted that anatase is not an equilibrium form of $TiO_2$ and, when heated in air to 600–900 °C, turns into rutile [15]. Thus, this study indicates that anatase nanoparticles in a glassy carbon matrix are stable even after exposure to 970 °C for an hour. Holding composites at that temperature might be essential for their performance. For example, it was reported that a photoelectrochemical experiment with a single crystal wafer of rutile treated at 700 °C under $10^{-4}$ torr for 4 h increased the conductivity of the crystal [40]. Koo et al. [41] summarized that the partial reduction of $Ti^{4+}$ to $Ti^{3+}$, together with the intercalation of protons and/or the formation of oxygen vacancies, could be a feasible self-doping that can be performed by the hydrogenation, chemical, and electrochemical reduction of $TiO_2$. The electric resistivity of both our composites measured by the two-point method is just 1.3–1.8 times larger than that of pure glassy carbon. It is known that $TiO_2$ nanoparticles have a lower conductivity in comparison with $TiO_2$ films [42]; however, these nanoparticles possess excellent compatibility and high hydrophilicity (necessary for the immobilization of biomolecules), which is mandatory for the construction and preparation of sensors. Several approaches can be used to improve their conductivity [43,44]. One of the best substrates for this purpose is carbon-based materials, such as amorphous ones, and they are used for the preparation of composites and the construction of electrochemical

sensors for the detection of various organic compounds [45]. This is based on their high electrical conductivity, good biocompatibility, and excellent electrochemical properties. Based on these studies, we incorporated $TiO_2$ nanoparticles into homemade glassy carbon to investigate its electrocatalytic properties for the detection of environmental hazards.

Thermal analysis of samples by heating at 10 °C/min to 1000 °C in air resulted in the burning off of the glassy carbon matrix (Figure 3A,B), although its pieces retained the initial shape of the composite fragments. Nanosized anatase particles passed into the well-crystallized 100–1500 nm rutile crystals (Figure 3C,D). From the mass loss, it was confirmed that the $TiO_2$ content in the composites was: 66.5 wt.% $TiO_2$ in IS24 and 25.5 wt.% $TiO_2$ in IS29. It should be noted that burning started at relatively low temperatures of 350–370 °C, which also might be the issue of the catalysis of this reaction by $TiO_2$.

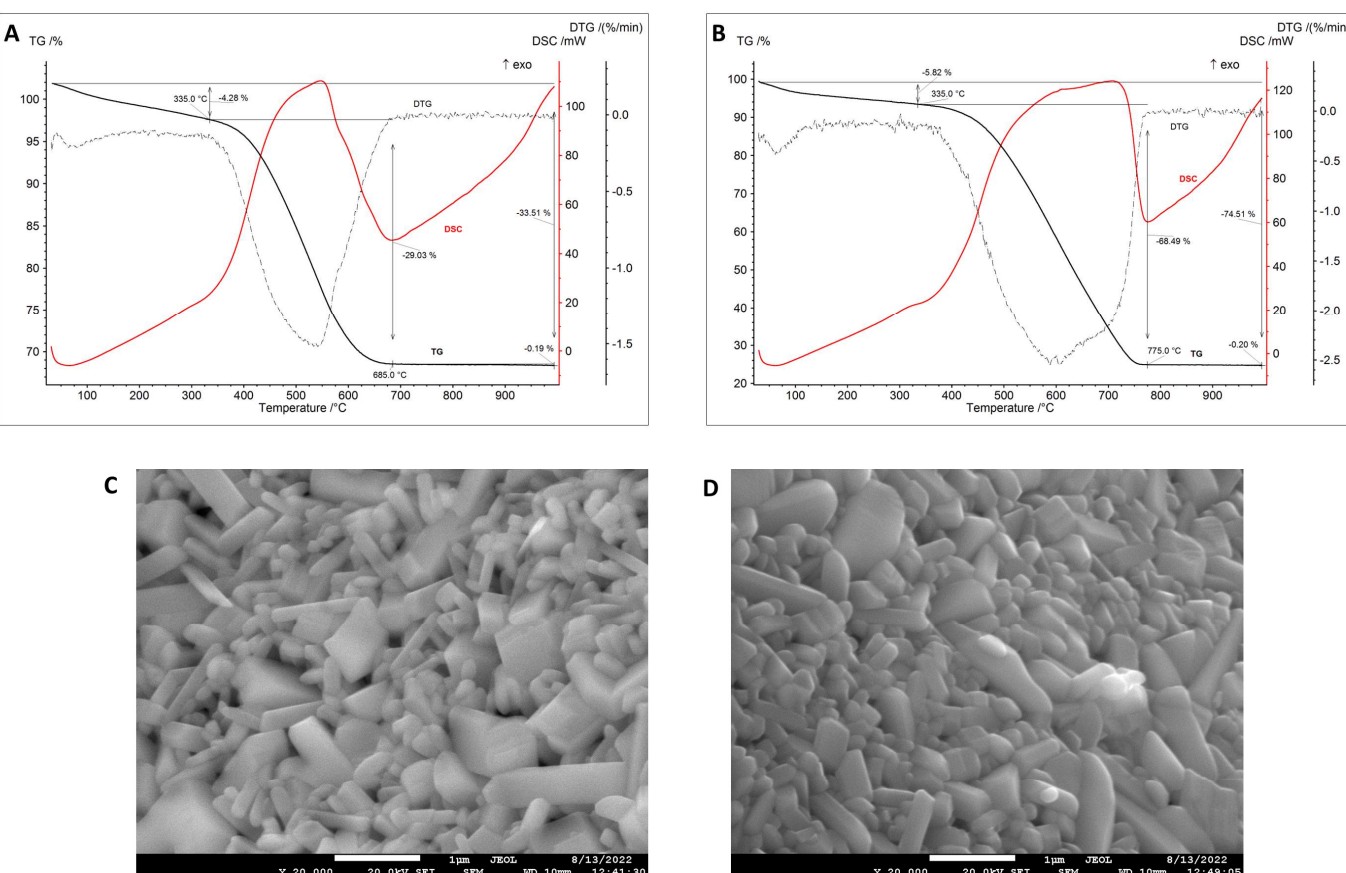

**Figure 3.** (**A**,**B**) TG-DSC curves in air of IS24 and IS29, respectively; (**C**,**D**) $TiO_2$ remained after TG-DSC in air of IS24 and IS29, respectively.

## 2.2. Electrochemical Characterization of the Electrodes

EIS was employed for the electrochemical characterization of the electrode surface— bare CPE and electrodes modified with synthesized nanocomposites IS24 and IS29 consisting of domestic glassy carbon and different crystalline phases and their amount of $TiO_2$. The EIS plots are shown in Figure 4A, which were recorded in 0.1 M KCl solution containing 5 mM $[Fe(CN)_6]^{3-/4-}$, at a potential of 0.3 V, a frequency ranging from 0.01 to 100,000 Hz, and an amplitude of 5 mV. The EIS plot exhibits a semicircle along with a straight line, with the semicircle indicating the electron charge transfer process and the straight line indicating the diffusion process. According to the EIS study, the Nyquist plot for the bare CPE had a higher Rct (22,580 Ω) value than the modified electrodes (Figure 4B), which confirms the bare CPE has less electron transfer capability due to its poor active sites. Regarding modified electrodes, it can be noticed that the electrode modified with nanocomposite IS29 possessed a lower Rct value (17,820 Ω) compared with the other modified electrode

(Rct value 20,450 Ω). This is probably due to the smaller crystalline size of sample IS29, which correlates with the effective surface area. In addition, it was shown that sample IS24 consisted of 6 wt.% rutile phases, which further contributes to the smaller active surface area.

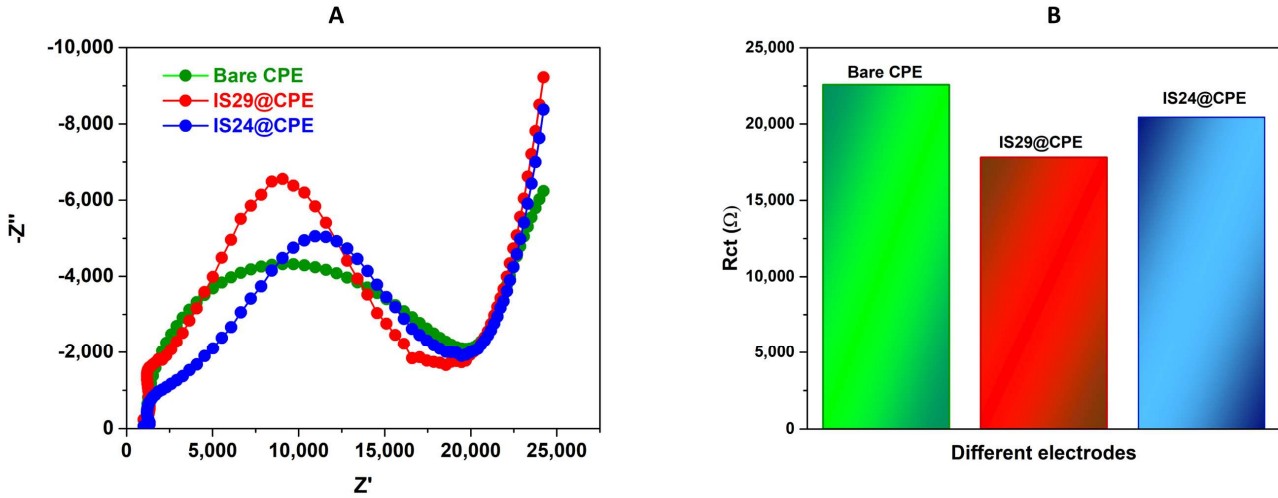

**Figure 4.** (**A**) EIS spectra of bare CPE and modified electrodes (IS29@CPE; IS24@CPE); (**B**) corresponding bar diagram of different electrodes vs. Rct.

In order to further investigate the electrocatalytic properties of the obtained electrodes, cyclic voltammograms of a solution containing 5 mM $[Fe(CN)_6]^{4-/3-}$ in 0.1 M KCl was recorded. Cyclic voltammograms were recorded at two scan rates, 20 and 50 mV/s, and are presented in Figures 5A and 5B, respectively. As can be observed, regardless of the scan rate, the behaviors of the electrodes were in the same order. However, with an increase in the scan rate from 20 to 50 mV/s, differences started to be more visible, especially in the peak-to-peak separation and system reversibility, indicating a higher catalytic potential of the materials in comparison with the unmodified electrode. All three electrodes showed a pair of two oval-shaped and well-defined redox peaks; oxidation peaks appeared around a potential of 0.3 V, and corresponding reduction peaks at a potential of around 0.06 V. The highest current intensity was achieved using the CPE modified with the IS29 composite (IS29@CPE). Once again, results showed that when the proportion of the rutile phase of $TiO_2$ in the composite increased, which was the case with the composite IS24, a slight decrease in current intensity was observed. This is probably a consequence of active surface area loss, which further led to limitations in charge transport in the nanocomposites and the poorest electrochemical response [46]. In order to further investigate the ability of the IS29 composite, we tested the effect of catalyst loading on the electrode structure. CV experiments, with the previously described conditions, were conducted using an electrode modified with 7.5, 10, 20, and 30% of the IS29 composite. These values, calculated on the amount of $TiO_2$, were 1.9, 2.5, 5, and 7.5%. Results for the measurements are summarized in Figure 5C. It is noted that the catalyst dosage upsurges were followed by enhanced electrode performance regarding mass transport, diffusion, and effective surface area. After an increase of over 2.5% in $TiO_2$ (10% of the composite), a reduction in $Fe^{2+/3+}$ redox currents was noted. This can be assigned to the increased amount of titanium dioxide at the electrode surface, and its low conductivity covers all the advantages of its excellent compatibility. Based on that, we selected an electrode with a 10% modifier for all further studies. Additional confirmation was obtained from the calculation of effective surface areas for the tested electrodes. The electroactive surface was calculated according to the Randles–Sevcik equation for the reversible electrode process: $A = I_a/(2.69 \times 10^5 \times A \times D^{1/2} \times n^{3/2} \times v^{1/2} \times C)$, where A is the electroactive area in $cm^2$, $I_a$ is the anodic current peak in A, D is the diffusion coefficient ($6.1 \times 10^{-6}$ $cm^2/s$ in the case of $[Fe(CN)_6]^{4-}$ in solution), n is the number of

electrons transferred in half-reaction (1 for $[Fe(CN)_6]^{4-}$), $\nu$ is the scan rate (0.05 V/s was used), and C is the concentration of $Fe(CN)_6^{4-}$ in M. Obtained for 7.5, 10, 20, and 30% modified electrodes were 2.3 mm$^2$, 3.8 mm$^2$, 3.1 mm$^2$, and 3.4 mm$^2$, respectively. The increase in the surface areas is excellent evidence that the modification amount is directly connected with the electrode characteristics and is sufficient evidence that the carbon paste electrode was successfully modified.

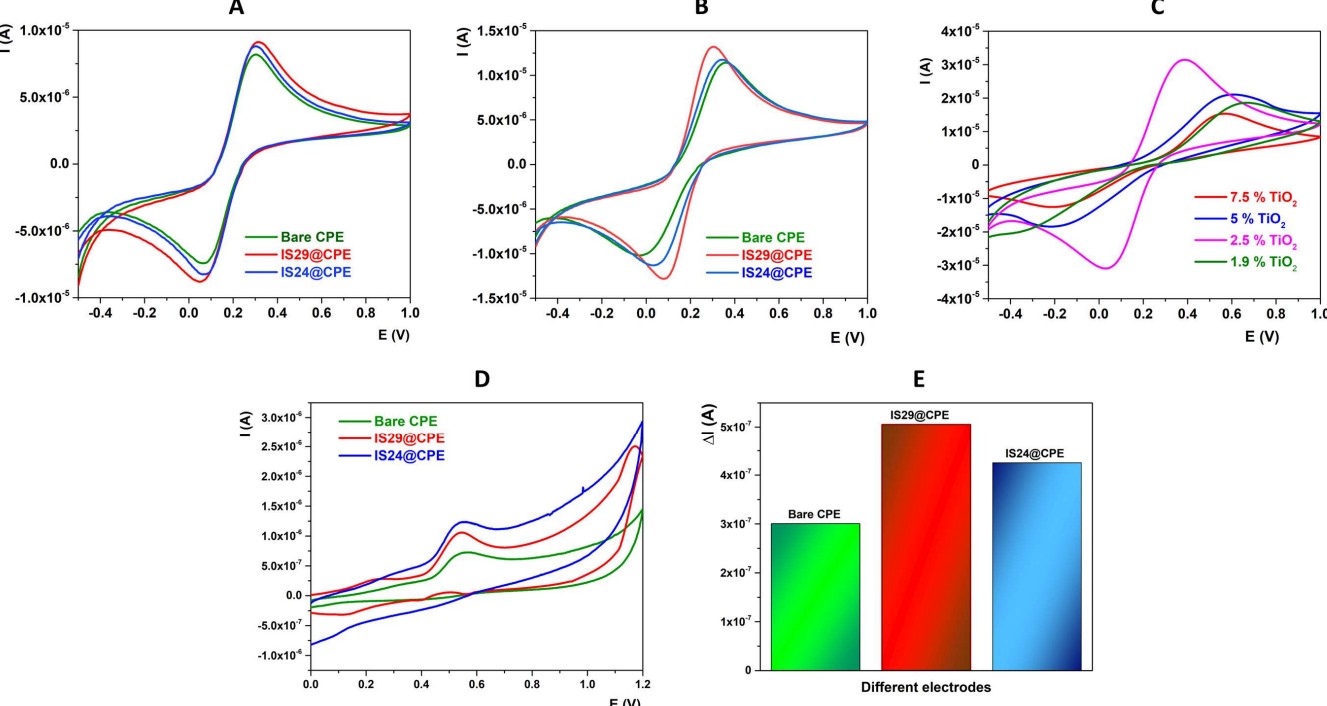

**Figure 5.** (**A,B**) CV probe for bare and modified electrodes (IS29@CPE; IS24@CPE) in the presence of $[Fe(CN)_6]^{3-/4-}$ medium with a scan rate 20 mV/s and 50 mV/s, respectively; (**C**) CV probe for electrode modified with composite consisting of 1.9, 2.5, 5, and 7.5% of TiO$_2$ (**D**); CV probe for bare and modified electrodes (IS29@CPE; IS24@CPE) in the presence of 50 µM triclosan in BRBS (pH 9) as a supporting electrolyte; (**E**) corresponding bar diagram of different electrodes vs. $\Delta$I.

The electrochemical sensing performance of the modified electrodes was estimated using 50 µM triclosan in BRBS (pH 9) as a supporting electrolyte with a potential ranging between 0.0 and +1.2 V at a scan rate of 50 mV/s. An analogous condition was carried out to the bare CPE in the presence of 50 µM triclosan. Based on obtained voltammograms (Figure 5D), it can be noticed that both modified electrodes possessed a better response for triclosan detection compared with the bare CPE electrode. The electrode modified with nanocomposite IS29 had the best electrochemical response toward triclosan, the best peak shape as well as the highest current (Figure 5E), so we chose this electrode for further development of an analytical method for the detection of this compound.

### 2.3. Optimization of Experimental Parameters

#### 2.3.1. Effect of the pH of the Supporting Electrolyte

Once we confirmed that our electrode possessed the best electrochemical characteristic for triclosan detection, we proceeded to develop an analytical procedure for the determination of triclosan. Firstly, we optimized the pH value of the supporting electrolyte. The voltametric oxidation of triclosan was examined in the pH range of 2–9 in 0.1 M BRBS (potential range from 0 to 1.2 V, scan rate 50 mV/s). At higher pHs of the supporting electrolyte, the residual current started to increase rapidly, so these pHs were not included in this study. Obtained voltammograms are given in Figure 6A. As can be noticed, the oxi-

dation peak potential shifted toward less positive values with an increase in the solution pH, indicating that protons are involved in the oxidation of triclosan. In addition, in solutions with medium pH values, the appearance of a second peak was observed. For both peaks, the potential was proportional to the solution pH in the examined range (Figure 6B). The regression equation demonstrated that equal numbers of protons and electrons are involved in the electron transfer process since the slopes of the curves were 56.5 and 54.8 mV/pH, respectively, which is very close to the theoretical value of 59 mV/pH for these processes. Calculating the slopes for pHs 4 and 5 in the Tafel region (the linear raising parts of the voltametric profile of triclosan recorded at a scan rate of 50 mV/s) for both processes, we obtained slopes values of 140 mV and 64 mV. These values are very close to the theoretical values for a one-electrode reaction (120 mV) and a two-electrode reaction (60 mV) [47]. Based on these calculations, we can assume that the electrocatalytic oxidation of triclosan at the proposed electrode is a two-step reaction (Figure 6D), which is in accordance with previous studies [48]. The oxidation mechanism for TCS oxidation is a two-step process. In the first step, involving one electron and one proton transfer, phenoxy radical formation occurs. This radical is stabilized with resonance and probably attacked by a water molecule–chemical oxidation mechanism. In the final step, a reversible formation of two different quinone products occurs with the involvement of two protons and two electrons. Finally, the highest signal current, as well as a good, defined peak shape, was achieved when the BRBS solution with pH 9 was used, so this solution was chosen for further experiments.

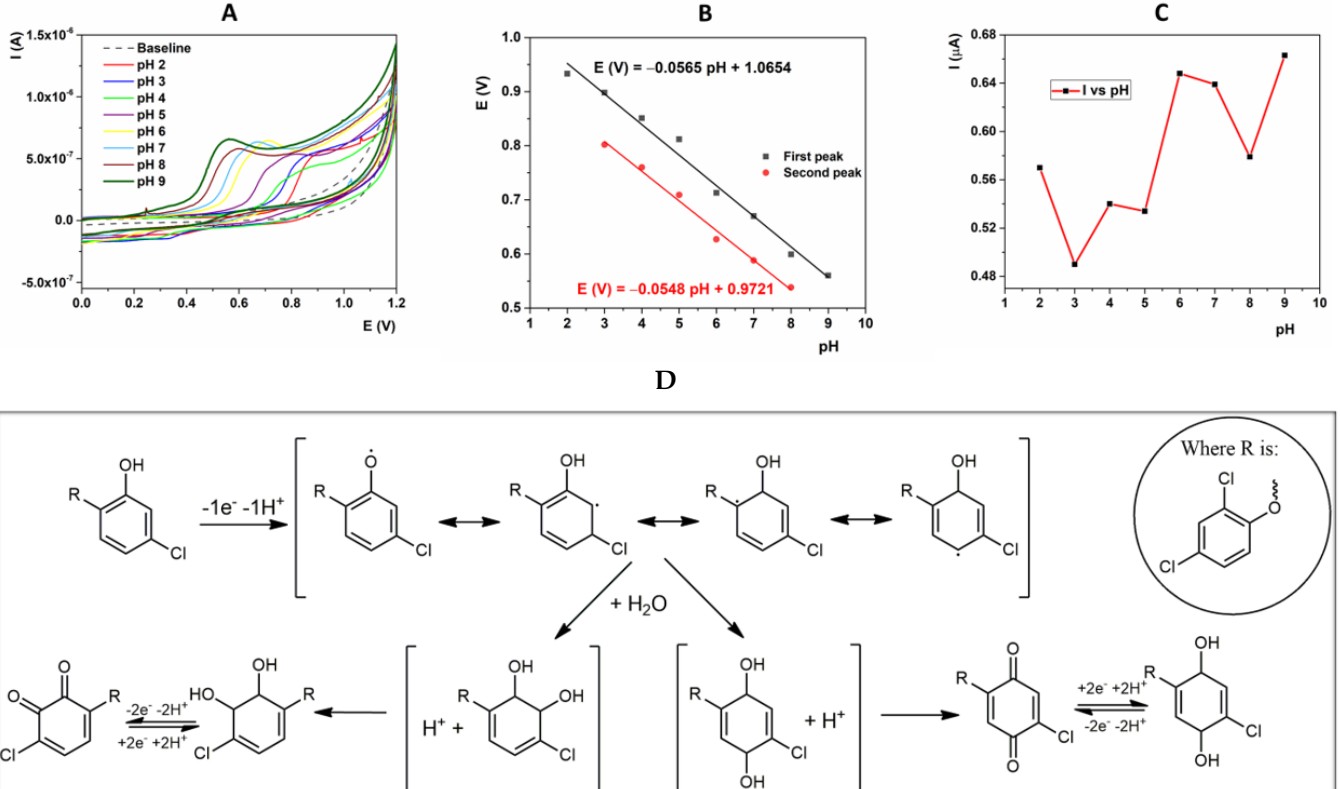

**Figure 6.** (**A**) CV profile of various pH (2–9) in the presence of BRBS containing 50 μM of TCS; (**B**) linearity diagram between potential vs. pH; (**C**) linearity diagram between current intensity vs. pH; (**D**) proposed oxidation mechanism for triclosan catalyzed at IS29@CPE.

2.3.2. Nature of the Electrochemical Reaction at the Interface Electrode/Testing Solution

The nature of the electrochemical reaction at the interface electrode/testing solution was examined using scan rate studies in the range of 10–100 mV/s (Figure 7) in BRBS pH 9 in the presence of 50 μM of triclosan. As shown in Figure 7A, raising the scan rate led to a shift in the oxidation peak toward more positive potential values with

a simultaneous increase in peak current intensity. The linear connection between the current intensity and scan rate is given in Figure 7B, and the corresponding linear equation is Ipa = $3.2254 \times 10^{-7} + 7.2952 \times 10^{-7}$ υ(mV/s) (R = 0.9882). This indicates that the electrochemical detection of TCS on the proposed modified electrode is a dominantly adsorption-controlled process.

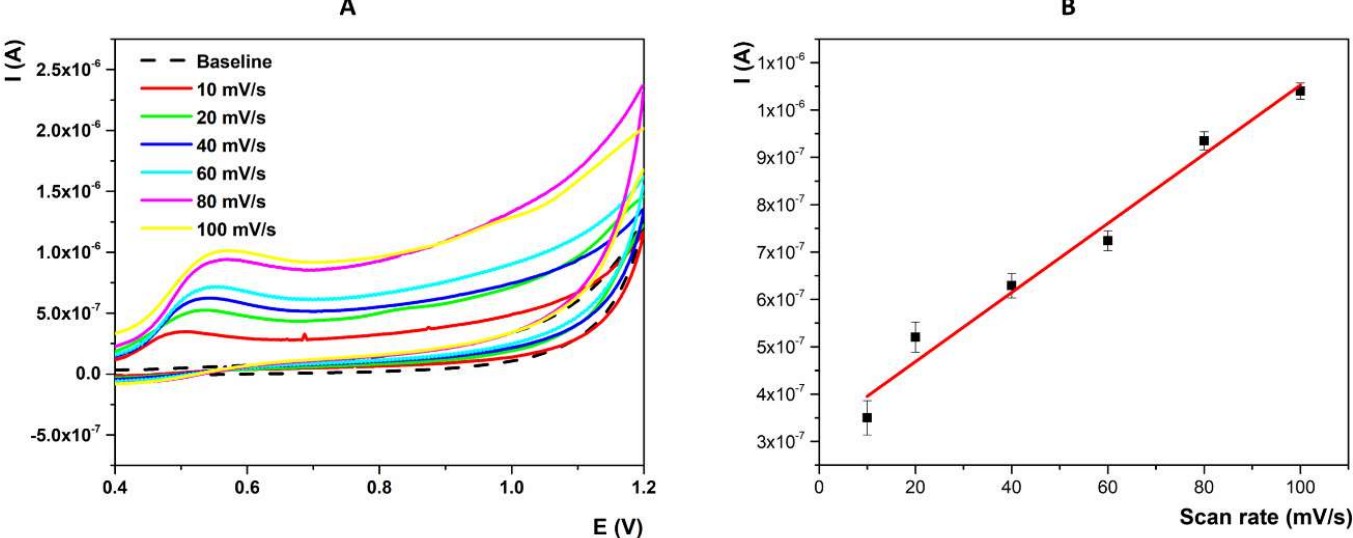

**Figure 7.** (**A**) CV curves of IS29@CPE at differing scan rates from 10 to 100 mV/s in a solution containing 50 μM of TCS in 0.1 M BRBS (pH 9.0); (**B**) corresponding calibration plot between peak current versus scan rate (mV/s).

### 2.4. Analytical Parameters of the Developed Method

A DPV study was carried out in order to determine the limit of detection (LOD) and the sensitivity of the proposed sensor with regard to the detection of TCS. The calibration curve was recorded in BRBS (pH 9) at a standard scan amplitude of 50 mV/s, a pulse time of 50 ms, and a potential increment of 4 mV. The concentrations of TCS added were in the range of 0.1 μM to 15 μM. Figure 8A illustrates the DPV signals of TCS from the lowest to the highest concentration. The relations between the peak current and the analyte concentration are shown in Figure 8B. It can be noticed the existence of two linearities such as a lower (up to 1 μM) and a higher concentration linearity. When it comes to low concentrations of TCS, the electrochemical process on the electrode surface is controlled mainly by diffusion, while at higher analyte concentrations, adsorption is the dominant process. As a result, two linear ranges were obtained. The lower linearity was founded to be from 0.1 to 1 μM with the corresponding linear equation: I ($\times 10^{-7}$ A) = 1.051 c (μM) + 0.414. The second linearity is described by the equation I (μA) = 0.161 c (μM) + 1.681. The limit of detection was calculated according to 3 Sa/b [49] as 0.07 μM based on the first linear range. In addition, amperometric i-t curves were recorded at an operation potential of 0.5 V and for 15 s. The increase in the TC concentration was followed by an increase in the current as a result of TCS oxidation, indicating that this outputting current is directly proportional to the TCS concentration. These results suggest that the proposed sensor can be further improved and the whole technology potentially transferred to the electrochemical disposable detection tool.

The repeatability of the developed IS29@CPE sensor was examined by recording five successive probes of 2 μM TCS standard solution. The obtained relative standard deviation (RSD) of 3.7% demonstrated a well-fabricated sensor with excellent repeatability. The reproducibility was estimated with five different electrodes, which were constructed independently using the proposed procedure. The RSD was 4.1% for the peak current measured in 2 μM TCS in BRBS (pH 9.0), which demonstrates the reliability of the fabrication procedure. This proves the proposed sensing platform shows remarkable sensitivity and

reproducibility. Moreover, the stability of the developed sensor was evaluated by recording 2 μM TCS standard solution after a 1-month period. The current decrease after this period was lower than 5%, which indicates the excellent stability of the sensor. In order to explore the stability of the sensor, a concentration of 2 μM of TCS was monitored for 1 month. Every 3 days, this concentration was measured using the prepared sensor. During this period, the electrode was stored under laboratory conditions. The relative standard deviation obtained from these measurements was 5%, and the final current decrease was lower than 3% from the initial value. This study indicates that the proposed preparation procedure and the developed sensor have good stability over the tested period.

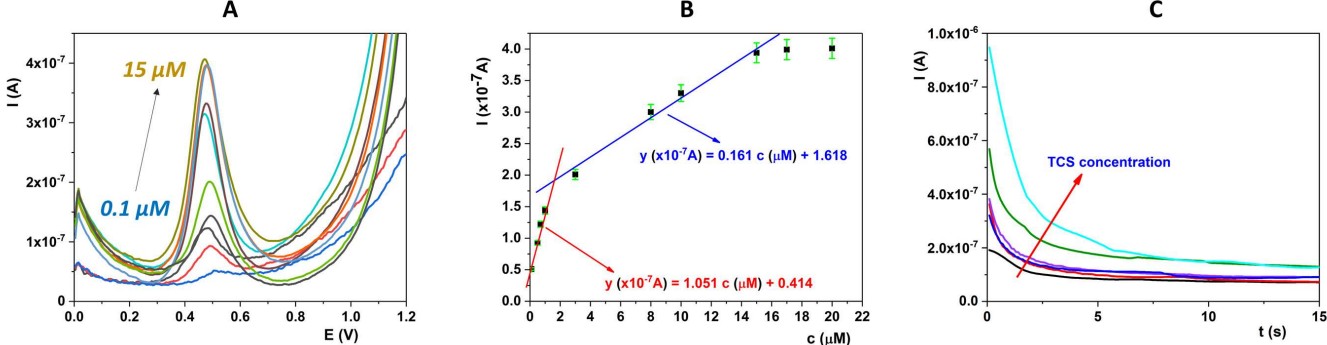

**Figure 8.** (**A**) DPV signal of IS29@CPE for TCS detection at the addition of 0.1–15 μM; (**B**) corresponding linear graph between peak current and concentration of TCS; (**C**) amperometric i-t curve for TCS different concentrations over proposed sensor.

All presented obtained results are comparable to those reported in the literature using different types of working electrodes, as can be seen in Table 1. Although there are developed sensors that have better analytical performance, such as detection limit and/or wider linear range, generally, the development of these sensors is much more complicated and time-consuming compared with the sensor proposed in this paper.

**Table 1.** Comparison of analytical parameters for TCS electrochemical detection employing different working electrodes.

| Electrode | Linearity Range (μM) | Detection Limit (μM) | References |
|---|---|---|---|
| IS29@CPE | 0.1–15 | 0.07 | This work |
| N-HC-modified GCE | 0.09 to 20 | 0.03 | [36] |
| LaFeO$_3$/Fe$_2$O$_3$@g-C$_3$N$_4$/SPE | 0.3–7 | 0.09 | [50] |
| AB/CNT/SB16 | 0.01–2 | 0.0062 | [51] |
| ITO/(Au-PIL/PDAC) | 10.0–60.0 | 0.098 | [52] |
| ZIF-11/activated carbon derived from rice husk | 0.1–8 | 0.076 | [53] |
| MIP-CQDs@HBNNS-NCs/GCE | $2 \times 10^{-3}$–100 | 0.005 | [54] |

**Abbreviation**: IS29, proposed composite consists of domestic glassy carbon and TiO$_2$ nanoparticles in anatase phase; CPE, carbon paste electrode; N-HC, N-doped hollow carbon; GCE, glassy carbon electrode; g-C$_3$N$_4$, graphitic carbon nitride; SPE, screen-printed electrode; AB, acetylene black; CNT, carbon nanotube; SB16, cetyl-sulphonyl betaine; ITO, indium-doped tin oxide; Au-PIL, Au nanoparticle-poly(ionic liquid); PDAC, poly(diallyl dimethyl ammonium) hydrochloride; ZIF-11, zeolite imidazolate framework-11; MIP, molecularly imprinted polymer; CQD, carbon quantum dot; HBNNS-NCs, hexagonal boron nitride nanosheet nanocomposites.

### 2.5. Interferences Studies

The interference study was performed to determine the selectivity of the proposed sensor for detecting TCS in the presence of possible cointerfering species, such as metal ions and common biological compounds. Metal ions commonly present in water (Ca$^{2+}$, Mg$^{2+}$, Na$^+$, K$^+$) have no effect on TCS determination. Some possible substances that are found in groundwater that may have an influence on triclosan determination, such as nitrites, phenols, and humic acids, were examined. Nitrites and humic acids did not affect the

determination of TCS because, in their presence (10 times higher concentration compared with the analyte), there was no significant change in the current intensity of the analyte peak (below 5%). Phenols, on the other hand, made it impossible to accurately determine the concentration of the analyte because their presence contributed to the increase in the TCS signal by 30% (Figure 9A). As one of the main sources of TCS, toothpaste, as indicated, for the possible application of the proposed method to TCS monitoring in such kinds of samples, we tested starch, maltose, glucose, fructose, and mannitol as potential interfering compounds (Figure 9B). In the presence of these compounds, the proposed sensor possessed excellent selectivity, as the current differences without and with the presence of these compounds were lower than 5%, indicating that the developed approach can be successfully applied to TCS monitoring in such samples.

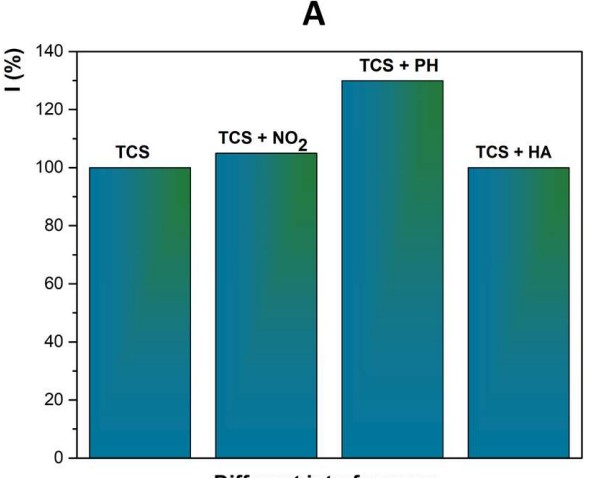 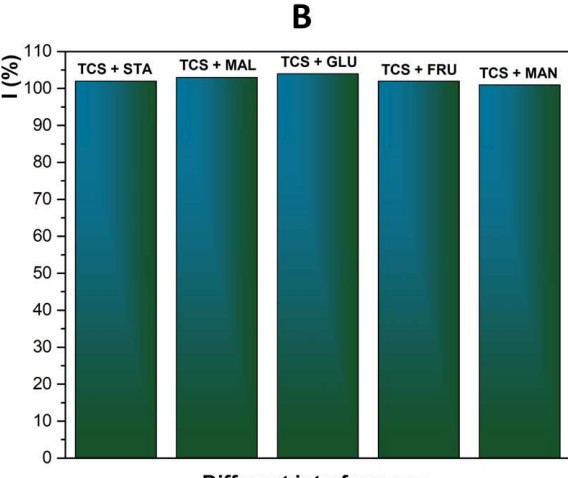

**Figure 9.** Interference studies for TCS determination (**A**) for the real-time water samples; (**B**) real-time tooth-paste samples.

### 2.6. Real Sample

For real sample analysis, three samples of groundwater were examined. All measurements were performed under optimized conditions for this sensing platform. All real samples did not show the relevant response for TCS, indicating those samples were not contaminated by this compound. To confirm that our proposed method can be used for real sample analysis, groundwater samples were spiked with a known amount of TCS, and recovery tests were conducted. The obtained results and calculated recoveries are presented in Table 2. These results imply that the proposed sensor may be successfully adapted to detect and monitor TCS concentration in groundwater.

**Table 2.** Real sample analysis.

| Sample No. | Found (µM) | Added (µM) | Found (µM) | Recovery (%) |
| --- | --- | --- | --- | --- |
| 1 | 0.00 | 1.00 | 1.05 | 105 |
| 2 | 0.00 | 3.00 | 3.08 | 103 |
| 3 | 0.00 | 5.00 | 4.93 | 99 |

## 3. Materials and Methods

### 3.1. Chemicals and Apparatus

All chemicals were of analytical grade and were used as supplied without any further purification. Boric acid, acetic acid, phosphoric acid, sodium hydroxide, potassium chloride (KCl), potassium ferricyanide ($K_3[Fe(CN)_6]$), and potassium ferrocyanide ($K_4[Fe(CN)_6]$) were all supplied by Merck (Darmstadt, Germany).

Stock solution of tested compound ($1 \times 10^{-3}$ M) was prepared by dissolving proper amount of TCS in ultrapure water and stored in a refrigerator at 4–6 °C. Working solutions of triclosan were freshly prepared from a stock solution prior to every experiment. Calibration solutions were prepared from the stock solution by appropriate dilution with supporting electrolyte. As supporting electrolyte Britton–Robinson buffer solutions (BRBS) (40 mM each of phosphoric acid, acetic acid, and boric acid) were used. The pH values of BR buffer were adjusted with sodium hydroxide (0.2 M). pH meter (Orion 1230, Thermo Fisher Scientific, Waltham, Massachusetts, USA) equipped with combined glass electrode (model Orion 9165BNWP, Thermo Fisher Scientific, Waltham, Massachusetts, USA) was used for all pH measurements.

All electrochemical measurements were performed using a potentiostat/galvanostat CHI 760b (CH Instruments, Inc., Austin, TE, USA). Electrochemical measurements were conducted in three-electrode glass cells (total volume of 25 mL) with an Ag/AgCl electrode (3M KCl) as reference electrode and Pt wire as counter electrode. Each potential reported in this paper is given against Ag/AgCl/3M KCl electrode at a laboratory temperature of $25 \pm 1$ °C.

Visualization of sample morphology was performed with transmission electron microscope Jeol JEM-2100F (JEOL, Tokyo, Japan) and scanning electron microscope Jeol JSM-7001F (JEOL, Tokyo, Japan). Crystalline phases were determined with powder X-ray diffractometer Rigaku Ultima IV (Cu radiation) (Tokyo, Japan). Thermal analysis was performed on simultaneous thermal analyzer Netzsch STA 449 F1 Jupiter (Selb, Germany).

### 3.2. Preparation of TiO$_2$/C

For the synthesis of glassy carbon composite, reagent-grade furfuryl alcohol (FA), tetrabutyltitanate Ti(OBu)$_4$ (TBT), and nonionic surfactant polyethyleneglycol (10) ether of isooctylphenol (OP-10) were used, as described before [21,22]. Solution of reagent-grade p-toluenesulfonic acid in n-butyl alcohol (36 wt.% of acid) was used as catalyst for FA polycondensation. It is an essential feature of described procedure that water resulting from polycondensation of FA is generated in situ with reagent that reacts with TBT to form hydrated titania nanoparticles.

Sample IS24 was prepared by mixing 2.041 g of FA, 4.065 g of TBT, 4.099 g of OP-10, and 0.20 mL of toluenesulfonic acid solution in butyl alcohol. Solution for sample IS29 contained 3.937 g of FA, 2.025 g of TBT, 4.113 g of OP-10, and 0.20 mL of toluenesulfonic acid in butyl alcohol. To accomplish polymerization, hydrolysis, and drying, the obtained homogenous solutions were heated at 50, 70, 90, 110, and 150 °C for 2 days at each temperature, resulting in solid samples. Then, they were heated in a container filled with soot at a rate of 50 C/h to 970 °C and calcined at that temperature for 1 h. This method starts from clear solution of reagents at wide range of concentrations, approving anatase nanoparticle formation and its uniform distribution in carbon matrix. It is suitable for bulk composite production, with high electric conductivity and strong mechanical bond of particles to matrix.

### 3.3. Preparation of Working Electrodes

An unmodified glassy carbon paste electrode was prepared by mixing 80% of glassy carbon powder and 20% of paraffin oil in a mortar for approximately 10 min to form the homogeneous glassy carbon paste. The modified glassy carbon paste electrodes were prepared by mixing unmodified paste with synthesized composite IS24 and IS29, respectively. The amount of the composites was 10% calculated on powder. After the mixture homogenization, the working electrode was filled with the paste. Subsequently, the electrode surface was polished with paper, washed with DI water, and directly used for measurements without pretreatment.

## 4. Conclusions

In this study, we investigated the electrocatalytic properties of homely synthesized titanium-dioxide-based nanomaterials regarding their possibilities to detect TCS. The

morphology of the materials was studied using XRD, TEM, and TGA analysis, while the electrocatalytic characteristics were studied with impedance spectroscopy and cyclic voltammetry. The obtained results proposed the prepared materials as excellent candidates for application in electroanalysis. Detailed parameters optimization and the estimated modification process indicate that the synergetic effect of the highly conductible glassy carbon and the biocompatibility properties of $TiO_2$ are excellent materials for electrochemical sensing. In this sense, a novel approach was developed for the sensitive, selective, and accurate determination of triclosan with a submicromolar detection limit. Excellent repeatability, reproducibility, and the long lifetime of the prepared sensor were achieved. All the analytical properties of the method can be based on the fabricated homemade composite—improved effective surface area, small particle size, and low charge transfer resistance. Based on these facts, the successful application in a real-time sample analysis was shown, with recovery in the range of 99 to 105%. Thus, the prepared materials can serve as an excellent basis for further investigation in this field and potential technology transfer, as the selected composite acts as a promising electrocatalyst for the electrochemical determination of triclosan.

**Author Contributions:** Conceptualization, V.S., D.M., G.M.R. and D.A.Z.; methodology, D.S.T., D.A.U. and V.V.A.; formal analysis, V.S., D.S.T. and D.A.Z.; investigation, D.M.S., D.S.T. and D.A.Z.; resources, V.V.A. and D.M.S.; writing—original draft preparation, V.S., D.A.Z. and D.M.S.; writing—review and editing, D.M.S.; supervision, D.M.S.; project administration, V.V.A. All authors have read and agreed to the published version of the manuscript.

**Funding:** This work was supported by the Ministry of Science and Higher Education of the Russian Federation (agreement no. 075-15-2022-1135), South Ural State University, the Ministry of Education, Science and Technological Development of the Republic of Serbia (contract number: 451-03-68/2022-14/200168 and 451-03-9/2021-14/200026), and the EUREKA project call E!13303.

**Data Availability Statement:** The study did not report any data.

**Conflicts of Interest:** The authors declare that they have no known competing financial interest or personal relationships that could have appeared to influence the work reported in this paper.

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
