# Peer review of "Synthesis and Application of Domestic Glassy Carbon TiO2 Nanocomposite for Electrocatalytic Triclosan Detection"

_catalysts, doi:10.3390/catal12121571_

Round 1

Reviewer 1 Report

The manuscript presents a new voltammetric sensor for the determination of triclosan concentration in ground water. Although the problem that the authors tried to solve is important one, I cannot recommend this paper to publication in the present form due to numerous scientific errors and inaccuracies.

1.     It seems to me that the methodological part of the manuscript should be given above the presentation of results.

2.     The authors noted that TiO2 possesses semiconductor properties even in the Abstract. However, the electrical resistance of titanium dioxide is rather high due to a large energy gap. In fact, TiO2 is semiconductor only in the thin films (see, for instance, A.Yildiz, S.B.Lisesivdin, M.Kasap, D.Mardare, Electrical properties of TiO2 thin films, Journal of Non-Crystalline Solids, V. 354, Issues 45–46, 4944-4947 (doi: 10.1016/j.jnoncrysol.2008.07.009). The electrical conductivity of TiO2 can be significantly increased by its doping. Since electrical conductivity is essential for any electrode material, the reliable data on electrical conductivity of C/TiO2 composites synthesized by the authors should be given.

3.     P.3, lines 113-114 “TiO2 nanocomposite has 2.5 higher effectivity of phenol electrocatalytic oxidation…”. Which composite? In comparison with what?

4.     Indexing reflexes in the diffraction pattern (Fig. 2) should be given. Which reflexes are used for the calculation of the coherent scattering region?

5.     The equivalent circuit used for processing EIS data should mandatory be given.

6.     A pair of peaks in cyclic volammograms (Fig. 5A,B) corresponds to Fe(CN)63-/Fe(CN)64- redox transition. It is well known that in the concentration range studied by the authors this transition is reversible one and the current in the peak can be calculated by the Randles-Sevchik equation. The rate of this reaction is limited only by mass transport to the electrode surface. Under such conditions, one cannot discuss about electrocatalytic properties of the electrode, number of the active centers on its surface, etc. Only characteristics of mass transport can be found from voltammetric dependences.

7.     The scheme presented by the authors in Fig. 6 D is nether proposed by the authors from their experimental studies nor discussed by them during the interpretation of the results. It seems foreign in the context of the manuscript.

8.     According to Fig. 8 B, the dependence of I vs c(triclosan), is non-linear in the concentration range of 3–16 mcM due to the adsorption of various species formed in the course of triclosan oxidation at the electrode surface. Therefore, the actual linearity range of the proposed method is rather narrow. It is completely unclear how to determine the concentration of triclosan in the range of 1–4 mcM.

I am sorry but I have to reject this material.

Author Response

Reviewer 1.

The manuscript presents a new voltammetric sensor for the determination of triclosan concentration in ground water. Although the problem that the authors tried to solve is important one, I cannot recommend this paper to publication in the present form due to numerous scientific errors and inaccuracies.

Thank you for your time and effort. We appreciate all comments and conclusions. In the revised version, we have accepted all valuable suggestions and tried to improve our manuscript. All changes are highlighted in the revised version.

  1. It seems to me that the methodological part of the manuscript should be given above the presentation of results.

- Thank you for the suggestion. However, according to the Catalysts journal template this is the order of the sections and we followed Instructions for authors. 

  1. The authors noted that TiO2 possesses semiconductor properties even in the Abstract. However, the electrical resistance of titanium dioxide is rather high due to a large energy gap. In fact, TiO2 is semiconductor only in the thin films (see, for instance, A.Yildiz, S.B.Lisesivdin, M.Kasap, D.Mardare, Electrical properties of TiO2 thin films, Journal of Non-Crystalline Solids, V. 354, Issues 45–46, 4944-4947 (doi: 10.1016/j.jnoncrysol.2008.07.009). The electrical conductivity of TiO2 can be significantly increased by its doping. Since electrical conductivity is essential for any electrode material, the reliable data on electrical conductivity of C/TiO2 composites synthesized by the authors should be given.

- Thank you for noticing that this information was missing. In the revised version, we have given the electrical resistivity for our composites and a corresponding comparison of our results with literature data, including the proposed paper.

The electric resistivity of our both composites measured by two-point method is just 1.3-1.8 times larger than that of pure glassy carbon. It is known that TiO2 nanoparticles have a lower conductivity in comparison with TiO2 films [38], however, these nanoparticles possess excellent compatibility and high hydrophilicity (necessary for the immobilization of biomolecules), which is mandatory for the construction and preparation of sensors. Several approaches can be used to improve their conductivity [39,40]. One of the best substrates for this purpose are carbon-based materials, such as amorphous one, and they are used for the preparation of composites and the construction of electrochemical sensors for the detection of various organic compounds [41]. This is based on their high electrical conductivity, good biocompatibility and excellent electrochemical properties. Based on these studies, we incorporated TiO2 nanoparticles into homemade glassy carbon to investigate its electrocatalytic properties towards the detection of environmental hazards.”      

  1. P.3, lines 113-114 “TiO2 nanocomposite has 2.5 higher effectivity of phenol electrocatalytic oxidation…”. Which composite? In comparison with what?

- Thank you for noticing this. This was not well defined. In the revised version we added appropriate clarification.

“TiO2/C nanocomposite with 0.37 wt.% of carbon has 2.5 times higher effectivity of phenol photocatalytic oxidation comparing to pure TiO2.”

  1. Indexing reflexes in the diffraction pattern (Fig. 2) should be given. Which reflexes are used for the calculation of the coherent scattering region?

- Thank you for your suggestions. Here are two numbers calculated from broadening of reflections on diffractogram: 6.8 and 7.0 nm. The first number is from Rietveld full profile refinement, the second – is from Debye formula for strongest 100% (101) reflection. They are close to each other. Much more important that SAXS and TEM are both in a good agreement with that numbers.

We are thinking that indexing is not needed on that figure, since due to broadening of reflections it is not possible to depict, say, (103), (004) and (112) reflections individually, since they form in fact single reflection near 38° 2Θ.

We provided additional explanation in the revised manuscript:

“Anatase average crystallite size calculated from broadening of reflections on diffractogram is 6.8-7.0 nm (8.7 nm by SAXS) for IS24 and 4.4-4.7 nm (6.1 nm by SAXS) for IS29.”

  1. The equivalent circuit used for processing EIS data should mandatory be given.

- Thank you for this suggestion. This was omitted by technical mistake. In the revised version this was provided.

  1. A pair of peaks in cyclic volammograms (Fig. 5A,B) corresponds to Fe(CN)63-/Fe(CN)64- redox transition. It is well known that in the concentration range studied by the authors this transition is reversible one and the current in the peak can be calculated by the Randles-Sevchik equation. The rate of this reaction is limited only by mass transport to the electrode surface. Under such conditions, one cannot discuss about electrocatalytic properties of the electrode, number of the active centers on its surface, etc. Only characteristics of mass transport can be found from voltammetric dependences.

- Thank you for noticing this. We adopt this text. In the revised version we use these measurements to provide calculations of effective surface areas for the all modification of the electrodes. This is added in the text and highlighted.

“In order to further investigate the ability of the IS29 composite we tested effect of catalyst loading at the electrode structure. CV experiments, with the previously described conditions, were done using the electrode modified with 7.5, 10, 20 and 30 % of the IS29 composite. These values, calculated on the amount to TiO2 are 1.9, 2.5, 5 and 7.5 %. Results for measurements are summarized in the Figure 5 C. It is noted that catalysts dosage upsurges are followed with the enhanced electrode performances, regarding mass transport, diffusion and effective surface area. After increase over 2.5 % of TiO2 (10 % of composite) reduction in Fe2+/3+ redox currents are noted. This can be assigned to the increased amount of titanium dioxide at the electrode surface and its low conductivity covers all the advantages of its excellent compatibility. Based on that, we select electrode with 10 % of modifier for all further studies. Additional confirmation was obtained from the calculation of effective surface areas for the tested electrodes. The electroactive surface was calculated according to Randles-Sevcik equation for reversible electrode process: A = Ia / (2.69 * 105 * A x D1/2 * n3/2 * ν1/2 x C), where A is the electroactive area in cm2, Ia is anodic current peak in A, D is the diffusion coefficient (6.1 ∙ 10−6 cm2/s in case of [Fe(CN)6]4− in solution), n is the number of electrons transferred in half-reaction (1 for [Fe(CN)6]4−), ν is scan rate (0.05 V/s was used) and C is concentration of Fe(CN)6]4−) in M. Obtained for 7.5, 10, 20 and 30 % modified electrodes were 2.3 mm2, 3.8 mm2, 3.1 mm2 and 3.4 mm2, respectively. The increase in the surface areas is excellent evidence that modification amount is directly connected with the electrode characteristics and sufficient evidence that carbon paste electrode was successfully modified.                                           

  1. The scheme presented by the authors in Fig. 6 D is nether proposed by the authors from their experimental studies nor discussed by them during the interpretation of the results. It seems foreign in the context of the manuscript.

- Thank you for your comment. However, we think that these reactions are confirmed from our measurements. We calculated number of electrons and protons and based on the literature search and our results we proposed oxidation mechanism. Based on that we think that mentioned Figure 6D. should stay in the manuscript.   

  1. According to Fig. 8 B, the dependence of I vs c(triclosan), is non-linear in the concentration range of 3–16 mcM due to the adsorption of various species formed in the course of triclosan oxidation at the electrode surface. Therefore, the actual linearity range of the proposed method is rather narrow. It is completely unclear how to determine the concentration of triclosan in the range of 1–4 mcM.

- Thank you for your suggestion. We fully agree with this. This is commented from reviewer 2. In the revised version we provided additional measurements and provided new Figure 8, as it is suggested from reviewers.

“A DPV study was carried out in order to determine the limit of detection (LOD) and the sensitivity of proposed sensor with regard to the detection of TCS. Calibration curve was recorded in BRBS (pH 9) at a standard scan amplitude of 50 mV/s, pulse time 50 ms and potential increment of 4 mV. The concentrations of TCS added were in the range from 0.1 µM to 15 µM. Figure 8A illustrates the DPV signals of TCS from lowest to highest concentration. The relations between peak current and analyte concentration are shown in Figure 8B. It can be noticed the existence of two linearities such as lower (up to 1 µM) and higher concentration linearity. When it comes to low concentrations of TCS, the electrochemical process on the electrode surface is con-trolled mainly by diffusion, while at higher analyte concentrations, adsorption is the dominant process. As the result, two linear ranges were obtained. The lower linearity was founded to be from 0.1 to 1 µM, with corresponding linear equation I (x 10-7 A) = 1.051 c (µM) + 0.414. Second linearity are described by equation I (x 10-7 A) = 0.161 c (µM) + 1.618. The limit of detection was calculated according to the 3 Sa/b [41] as 0.07 µM, based on first linear range. In addition, amperometric i-t curves were recorder at the operation potential of 0.5 V and for 15 s. The increase in the TC concentration is followed with the increase of current as a result of TCS oxidation, indicating that this outputting current is directly proportional to the TCS concentration. These results suggest that proposed sensor can be further improved and the whole technology potentially transferred to the electrochemical disposable detection tool. “    

Figure 8. A) DPV signal of IS29@CPE towards TCS detection at the addition of 0.1–15 µM B) Corresponding linear graph between peak current and concentration of TCS; C) Amperometric i-t curved for TCS different concentrations oved proposed sensor.

Reviewer 2 Report

In this report, the authors prepared modified carbon paste electrode with domestic carbon material enhanced with TiO2 nanoparticles and characterized it with TEM, SEM, XRD and EIS. Further, the authors used this electrode to detect triclosan and obtained good results. I recommend its publication after minor revision.  

1.      The active area of the prepared electrode should be confirmed;

2.      The effect of the amount of TiO2 in the modified electrode on the catalytic activity should be investigated;

3.      The mechanism of the modified electrode should be discussed;

4.      The gotten results should be compared to that reported in the literature.

5.      DPV curves are very ugly. The authors should repeat this experiment;

6.      The i-t curves should be added;

7.      The stability of the prepared electrode should be evaluated;

8.      Interferences studies should be extended. The authors need to find some substances with the same oxidation potentials and test if the prepared electrode can electrocatalyze their oxidation to provide the actual selectively.   

Author Response

Reviewer 2.

In this report, the authors prepared modified carbon paste electrode with domestic carbon material enhanced with TiO2 nanoparticles and characterized it with TEM, SEM, XRD and EIS. Further, the authors used this electrode to detect triclosan and obtained good results. I recommend its publication after minor revision.

We are grateful to the reviewer for its time and efforts and positive comments of our work. We revised manuscript according to these comments and we hope that this improvement, based on reviewer comments will significantly improve final version of our work.

  1. The active area of the prepared electrode should be confirmed;

- Thank you for this suggestion. In the revised version we provided active surface areas for all electrodes tested in this work. According to the reviewers comments (reviewer 2, comment 2; and reviewer 3, comment 4) we tested additional electrodes, with different amounts of composite in the carbon paste electrode. For all these steps we calculated effective surface areas. This is added in the revised version and highlighted.

“In order to further investigate the ability of the IS29 composite we tested effect of catalyst loading it the electrode structure. CV experiments, with the previously described conditions, were done using the electrode modified with 7.5, 10, 20 and 30 % of the IS29 composite. These values, calculated on the amount to TiO2 are 1.9, 2.5, 5 and 7.5 %. Results for measurements are summarized in the Figure 4 C. It is noted that catalysts dosage upsurges is followed with the enhanced electrode performances, regarding mass transport, diffusion and effective surface area. After increase over 2.5 % of TiO2 (10 % of composite) reduction in Fe2+/3+ redox currents is noted. This can be assigned to the increased amount of titanium dioxide at the electrode surface and its low conductivity covers all the advantages of its excellent compatibility. Based on that, we select electrode with 10 % of modifier for all further studies. Additional confirmation was obtained from the calculation of effective surface areas for the tested electrodes. The electroactive surface was calculated according to Randles-Sevcik equation for reversible electrode process: A = Ia / (2.69 * 105 * A x D1/2 * n3/2 * ν1/2 x C), where A is the electroactive area in cm2, Ia is anodic current peak in A, D is the diffusion coefficient (6.1 ∙ 10−6 cm2/s in case of [Fe(CN)6]4− in solution), n is the number of electrons transferred in half-reaction (1 for [Fe(CN)6]4−), ν is scan rate (0.05 V/s was used) and C is concentration of Fe(CN)6]4−) in M. Obtained for 7.5, 10, 20 and 30 % modified electrodes were 2.3 mm2, 3.8 mm2, 3.1 mm2 and 3.4 mm2, respectively. The increase in the surface areas is excellent evidence that modification amount is directly connected with the electrode characteristics and sufficient evidence that carbon paste electrode was successfully modified.”                      

  1. The effect of the amount of TiO2in the modified electrode on the catalytic activity should be investigated;

      - Thank you suggesting this. This is also noticed from the reviewer 3. We tested two different amounts of TiO2 in the modified electrode. In the revised version we provided measurements for different loading amounts of modifier.  These results are now included in the revised version, new figures are provided and corresponding discussion.

“In order to further investigate the ability of the IS29 composite we tested effect of catalyst loading it the electrode structure. CV experiments, with the previously described conditions, were done using the electrode modified with 7.5, 10, 20 and 30 % of the IS29 composite. These values, calculated on the amount to TiO2 are 1.9, 2.5, 5 and 7.5 %. Results for measurements are summarized in the Figure 4 C. It is noted that catalysts dosage upsurges is followed with the enhanced electrode performances, regarding mass transport, diffusion and effective surface area. After increase over 2.5 % of TiO2 (10 % of composite) reduction in Fe2+/3+ redox currents is noted. This can be assigned to the increased amount of titanium dioxide at the electrode surface and its low conductivity covers all the advantages of its excellent compatibility. Based on that, we select electrode with 10 % of modifier for all further studies. Additional confirmation was obtained from the calculation of effective surface areas for the tested electrodes. The electroactive surface was calculated according to Randles-Sevcik equation for reversible electrode process: A = Ia / (2.69 * 105 * A x D1/2 * n3/2 * ν1/2 x C), where A is the electroactive area in cm2, Ia is anodic current peak in A, D is the diffusion coefficient (6.1 ∙ 10−6 cm2/s in case of [Fe(CN)6]4− in solution), n is the number of electrons transferred in half-reaction (1 for [Fe(CN)6]4−), ν is scan rate (0.05 V/s was used) and C is concentration of Fe(CN)6]4−) in M. Obtained for 7.5, 10, 20 and 30 % modified electrodes were 2.3 mm2, 3.8 mm2, 3.1 mm2 and 3.4 mm2, respectively. The increase in the surface areas is excellent evidence that modification amount is directly connected with the electrode characteristics and sufficient evidence that carbon paste electrode was successfully modified.”                      

  1. The mechanism of the modified electrode should be discussed;

- Thank you for your suggestion. In the revised version we explained redox mechanism for TCS over proposed sensor.

“The oxidation mechanism for TCS is two-step process. In the first step, involving one electron and one proton transfer, phenoxy radical formation occur. This radical is stabilized with resonance and probably attacked by water – chemical oxidation mechanism. In the final step an reversible formation of two different quinone products occurs with the involvement of two protons and two electrons.”

  1. The gotten results should be compared to that reported in the literature.

- Thank you for your comment. This is also suggested by other reviewers. In the revised version we provided new Table where obtained results in this study are compared with new literature data (2020-2023).  

“All presented obtained results are comparable to those reported in the literature using different types of working electrodes as it can be seen in Table 1. Although there are developed sensors that have better analytical performance, such as detection limit and/or wider linear range, generally the development of these sensors is much more complicated and time-consuming compared to the sensor proposed in this paper. 

Table 1. Comparison of analytical parameters for TCS electrochemical detection employing different working electrodes.

Electrode

Linearity range (µM)

Detection limit (µM)

References

IS29@CPE

0.1 - 15

0.07

This work

N-HC modified GCE

0.09 to 20

0.03

[36]

LaFeO3/Fe2O3@g-C3N4/SPE

0.3 - 7

0.09

[50]

AB/CNT/SB16

0.01 - 2

0.0062

[51]

ITO/(Au-PIL/PDAC)

10.0-60.0

0.098

[52]

ZIF-11/activated carbon derived from rice husk

0.1 - 8

0.076

[53]

MIP - CQDs@HBNNS-NCs/GCE

2 * 10-3 - 100

0.005

[54]

Abbreviation: IS29 (proposed composite consist of domestic glassy carbon and TiO2 nanoparticles in anatase phase); CPE (carbon paste electrode); N-HC (N-doped hollow carbon); GCE (glassy carbon electrode); g - C3N4 (graphitic carbon nitride); SPE (Screen printed electrode); AB (acetylene black); CNT (carbon nanotube); SB16 (cetyl-sulphonyl betaine); ITO (indium-doped tin oxide); Au-PIL (Au nanoparticle-poly(ionic liquid)); PDAC (poly(diallyl dimethyl ammonium) hydrochloride);  ZIF-11 (zeolite imidazolate framework-11); MIP (molecularly imprinted polymer); CQD (carbon quantum dot); HBNNS - NCs (hexagonal boron nitride nanosheets nanocomposites) “

  1. DPV curves are very ugly. The authors should repeat this experiment;

- Thank you for your suggestion. We fully agree with this. This is commented from other reviewer too. In the revised version we provided additional measurements and provided new Figure 8, as it is suggested from reviewers.

“A DPV study was carried out in order to determine the limit of detection (LOD) and the sensitivity of proposed sensor with regard to the detection of TCS. Calibration curve was recorded in BRBS (pH 9) at a standard scan amplitude of 50 mV/s, pulse time 50 ms and potential increment of 4 mV. The concentrations of TCS added were in the range from 0.1 µM to 15 µM. Figure 8A illustrates the DPV signals of TCS from lowest to highest concentration. The relations between peak current and analyte concentration are shown in Figure 8B. It can be noticed the existence of two linearities such as lower (up to 1 µM) and higher concentration linearity. When it comes to low concentrations of TCS, the electrochemical process on the electrode surface is con-trolled mainly by diffusion, while at higher analyte concentrations, adsorption is the dominant process. As the result, two linear ranges were obtained. The lower linearity was founded to be from 0.1 to 1 µM, with corresponding linear equation I (x 10-7 A) = 1.051 c (µM) + 0.414. Second linearity are described by equation I (x 10-7 A) = 0.161 c (µM) + 1.618. The limit of detection was calculated according to the 3 Sa/b [41] as 0.07 µM, based on first linear range. In addition, amperometric i-t curves were recorder at the operation potential of 0.5 V and for 15 s. The increase in the TC concentration is followed with the increase of current as a results of TCS oxidation, indicating that this outputting current is directly proportional to the TCS concentration. These results suggest that proposed sensor can be further improved and the whole technology potentially transferred to the electrochemical disposable detection tool. “    

Figure 8. A) DPV signal of IS29@CPE towards TCS detection at the addition of 0.1–15 µM B) Corresponding linear graph between peak current and concentration of TCS; C) Amperometric i-t curved for TCS different concentrations oved proposed sensor.

  1. The i-t curves should be added;

- Thank you for your comment. In the revised version we provided i-t curves for different concentration of TCS and appropriate discussion.

In addition, amperometric i-t curves were recorder at the operation potential of 0.5 V and for 15 s. The increase in the TC concentration is followed with the increase of current as a result of TCS oxidation, indicating that this outputting current is directly proportional to the TCS concentration. These results suggest that proposed sensor can be further improved and the whole technology potentially transferred to the electro-chemical disposable detection tool.     

  1. The stability of the prepared electrode should be evaluated;

- Thank you for this suggestion. In the revised version stability studies are provided.

“In order to explore the stability of the sensor concentration of 2 μM of TCS is monitored during one month. Every three days this concentration is measured using prepared sensor. During this period electrode was stored at the laboratory conditions. Relative standard deviation obtained from these measurements was 5 %, and final current decrease was lower than 3 %, from the initial value. This study indicates that proposed preparation procedure and developed sensor have good stability over the tested period.”   

  1. Interferences studiesshould be extended. The authors need to find some substances with the same oxidation potentials and test if the prepared electrode can electrocatalyze their oxidation to provide the actual selectively.

- Thank you for your suggestion. In accordance with this we extended our interferences studies. As one of the main sources of the TCS, the tooth paste can be indicated. Based on this, we tested additional ions are potential interferences as well as starch, maltose, glucose, fructose and mannitol. New figure is provided and appropriate discussion.

 “As one of the main sources of TCS tooth paste is indicated. For the possibly application of the proposed method for the TCS monitoring in such kind of the samples, we tested starch, maltose, glucose, fructose and mannitol as potential interfering compounds (Figure 9B). In the presence of these compounds, a proposed sensor possesses excellent selectivity, as the current differences without and with presence of these compounds was lower than 5 %, indicating that developed approach can be successfully applied for the TCS monitoring in such samples.”

Reviewer 3 Report

The authors demonstrated that glass carbon-TiO2 composite catalyst for electrochemical determination of triclosan. Physicochemical and electrochemical properties were studied by XRD, TEM, TGA, CV and impedance spectroscopies. However, the research design was not appropriate and quality information was missing. I might therefore not recommend the manuscript unless substantial improvement can be achieved. Below are some comments for the authors to consider and improve the quality of the paper.

1.     The novelty of the work should be established.

2.     Compare your results with the literature ones. Provide a tabular form of the literature performance results with the present work.

3.     Please provide a brief explanation of the experimental synthesis of IS24 and IS29.

4.     Control experiments (different ratios of TiO2 and glassy carbon) are required.

5.     How do you confirm carbon is present in the samples? Raman analysis is required.

6.     We found several mistakes (typos/grammatical errors). For example, Line 75  triclosan is abbreviated as “TCS” and some other places in the article are mentioned as “TSC” and so on.

7.     Authors should state the complete form of each abbreviation at its first appearance.

8.     Level of English is good however in a few places some syntax errors are present. In some places two or more words joined together that should be corrected.

9.     Conclusion should be revised and improved.

10.  The introduction section should be modified and improved through citing recent references (2021 and 2022) related studies and indicating the novelty of the study compared to the carried works. The following references should be added. Also added citing references regarding TiO2-based composites (Materials Research Bulletin 2018, 98, 314-321; Journal of Solid State Electrochemistry 2018, 22, 129-139; ChemistrySelect 2017, 2, 65-73; International Journal of Environmental Analytical Chemistry 2022, 1-19 (DOI: 10.1080/03067319.2022.2106860).).

11.  Check that all references are relevant to the contents of the manuscript.

Author Response

Reviewer 3.

The authors demonstrated that glass carbon-TiO2 composite catalyst for electrochemical determination of triclosan. Physicochemical and electrochemical properties were studied by XRD, TEM, TGA, CV and impedance spectroscopies. However, the research design was not appropriate and quality information was missing. I might therefore not recommend the manuscript unless substantial improvement can be achieved. Below are some comments for the authors to consider and improve the quality of the paper.

We are thankful to the reviewer for its time and efforts and positive comments of our work. We revised manuscript according to these comments and we hope that this improvement, based on reviewer comments will significantly improve final version of our work.

  1. The novelty of the work should be established.

- Thank you for your suggestion. In the revised version we improved section Introduction using regarding novelty of the work. Appropriate reference are provided.

- Currently, considerable attention is focused on optimizing the production of carbon material in order to achieve morphological changes that would lead to the creation of more porous surfaces and improvement of the mechanical and electrical properties of the electrode material. In this sense, we present the synthesis of domestic carbon material, enhanced with different amount of TiO2 nanoparticles, and utilized for fabrication of modified carbon paste electrode which is further used for TCS detection. The anatase is the only TiO2 phase in one composite sample (IS29) and is a dominant phase in another sample (IS24). The composite material with bulk glassy carbon matrix and anatase nanoparticles has not been explored before. The proposed method of composite production insures its excellent mechanical and galvanic contact between conductive inert matrix and catalytic nanoparticles that both are crucial for performance of composite in electrochemistry. With this in mind, the catalytic capabilities of the working electrodes were investigated depending on the main crystalline phase of TiO2 in the electrode modification material.

  1. Compare your results with the literature ones. Provide a tabular form of the literature performance results with the present work.

- Thank you for your comment. This is also suggested by other reviewers. In the revised version we provided new Table where obtained results in this study are compared with new literature data.

“All presented obtained results are comparable to those reported in the literature using different types of working electrodes as it can be seen in Table 1. Although there are developed sensors that have better analytical performance, such as detection limit and/or wider linear range, generally the development of these sensors is much more complicated and time-consuming compared to the sensor proposed in this paper. 

Table 1. Comparison of analytical parameters for TCS electrochemical detection employing different working electrodes.

Electrode

Linearity range (µM)

Detection limit (µM)

References

IS29@CPE

0.1 - 15

0.07

This work

N-HC modified GCE

0.09 to 20

0.03

[36]

LaFeO3/Fe2O3@g-C3N4/SPE

0.3 - 7

0.09

[50]

AB/CNT/SB16

0.01 - 2

0.0062

[51]

ITO/(Au-PIL/PDAC)

10.0-60.0

0.098

[52]

ZIF-11/activated carbon derived from rice husk

0.1 - 8

0.076

[53]

MIP - CQDs@HBNNS-NCs/GCE

2 * 10-3 - 100

0.005

[54]

Abbreviation: IS29 (proposed composite consist of domestic glassy carbon and TiO2 nanoparticles in anatase phase); CPE (carbon paste electrode); N-HC (N-doped hollow carbon); GCE (glassy carbon electrode); g - C3N4 (graphitic carbon nitride); SPE (Screen printed electrode); AB (acetylene black); CNT (carbon nanotube); SB16 (cetyl-sulphonyl betaine); ITO (indium-doped tin oxide); Au-PIL (Au nanoparticle-poly(ionic liquid)); PDAC (poly(diallyl dimethyl ammonium) hydrochloride);  ZIF-11 (zeolite imidazolate framework-11); MIP (molecularly imprinted polymer); CQD (carbon quantum dot); HBNNS - NCs (hexagonal boron nitride nanosheets nanocomposites) “

  1. Please provide a brief explanation of the experimental synthesis of IS24 and IS29.

-     Thank you for your comment. In the revised version, in the section Materials and Methods, we provided additional explanation of the synthesis procedures.

“For the synthesis of glassy carbon composite the reagent grade furfuryl alcohol (FA), tetrabutyltitanate Ti(OBu)4 (TBT) and nonionic surfactant polyethyleneglycol (10) ether ,of isooctylphenol (OP- 10) were used as described before [17,18]. Solution of reagent grade p-toluenesulfonic acid in n-butyl alcohol (36 wt. % of acid) was used as catalyst for FA polycondensation. It is essential feature of described procedure that water resulted from polycondensation of FA is a generated in situ reagent that react with TBT to form hydrated titania nanoparticles.

Sample IS24 was prepared by mixing 2.041 g of FA, 4.065 g of TBT, 4.099 g of OP-10 and 0.20 ml of toluenesulfonic acid solution in butyl alcohol. Solution for sample IS29 contained 3.937 g of FA, 2.025 g of TBT, 4.113 g of OP-10 and 0.20 ml of toluenesulfonic acid in butyl alcohol. To accomplish polymerization, hydrolysis and drying, the obtained homogenous solutions were heated at 50, 70, 90, 110 and 150 °Ð¡ for 2 days at each temperature resulting in solid samples. Then they were heated in a container filled with soot at a rate of 50 C/h to 970°C and calcined at that temperature for one hour. This method starts from clear solution of reagents at wide range of concentration, approving anatase nanoparticle formation and its uniform distribution in carbon matrix. It is suitable for bulk composite production, with high electric conductivity and strong mechanical bond of particles to matrix.”

  1. Control experiments (different ratios of TiO2 and glassy carbon) are required.

-     Thank you suggesting this. This is also noticed from the reviewer 3. We tested different loading amounts of TiO2 in the modified electrode. In the revised version we provided measurements for three more amounts. These results are now included in the revised version, new figures are provided and corresponding discussion.

“In order to further investigate the ability of the IS29 composite we tested effect of catalyst loading it the electrode structure. CV experiments, with the previously described conditions, were done using the electrode modified with 7.5, 10, 20 and 30 % of the IS29 composite. These values, calculated on the amount to TiO2 are 1.9, 2.5, 5 and 7.5 %. Results for measurements are summarized in the Figure 5 C. It is noted that catalysts dosage upsurges is followed with the enhanced electrode performances, regarding mass transport, diffusion and effective surface area. After increase over 2.5 % of TiO2 (10 % of composite) reduction in Fe2+/3+ redox currents are noted. This can be assigned to the increased amount of titanium dioxide at the electrode surface and its low conductivity covers all the advantages of its excellent compatibility. Based on that, we select electrode with 10 % of modifier for all further studies. Additional confirmation was obtained from the calculation of effective surface areas for the tested electrodes. The electroactive surface was calculated according to Randles-Sevcik equation for reversible electrode process: A = Ia / (2.69 * 105 * A x D1/2 * n3/2 * ν1/2 x C), where A is the electroactive area in cm2, Ia is anodic current peak in A, D is the diffusion coefficient (6.1 ∙ 10−6 cm2/s in case of [Fe(CN)6]4− in solution), n is the number of electrons transferred in half-reaction (1 for [Fe(CN)6]4−), ν is scan rate (0.05 V/s was used) and C is concentration of Fe(CN)6]4−) in M. Obtained for 7.5, 10, 20 and 30 % modified electrodes were 2.3 mm2, 3.8 mm2, 3.1 mm2 and 3.4 mm2, respectively. The increase in the surface areas is excellent evidence that modification amount is directly connected with the electrode characteristics and sufficient evidence that carbon paste electrode was successfully modified.”                      

  1. How do you confirm carbon is present in the samples? Raman analysis is required.

-     Thank you for your comments. This is not well explained in the text. In the revised version we provided clarification for this. Carbon confirmed qualitatively by EDX (Oxford X-max 80) installed on SEM. Quantitative carbon content was derived from thermal analysis. Glassy state of carbon is confirmed by powder X-Ray diffraction. All that is already included into manuscript.  Unfortunately, due to a malfunction of the apparatus, we are not able to do FTIR measurements, so we proved the presence of carbon by other techniques.

  1. We found several mistakes (typos/grammatical errors). For example, Line 75 triclosan is abbreviated as “TCS” and some other places in the article are mentioned as “TSC” and so on.

- Thank you for noticing this, we fix all the typos.

  1. Authors should state the complete form of each abbreviation at its first appearance.

- Thank you. These were technical mistakes. In the revised version we provided complete form of each abbreviation at its first appearance.

  1. Level of English is good however in a few places some syntax errors are present. In some places two or more words joined together that should be corrected.

- Thank you, we omit all the mistakes.

  1. Conclusion should be revised and improved.

- Thank you for your suggestion. In the revised version we improved our section Conclusion.

“In this study we investigate electrocatalytic properties of homely synthesized titanium dioxide-based nanomaterials regarding their possibilities to detect TCS. Morphology of the materials was studied using XRD, TEM and TGA analysis, while electrocatalytic characteristics were studied with impedance spectroscopy and cyclic voltammetry. Obtained results proposed prepared materials as excellent candidates for application in electroanalysis. Detailed parameters optimization and estimated modification process indicate that synergetic effect of highly conductible glassy carbon and biocompatibility properties of TiO2 are excellent materials for electrochemical sensing. In this sense, novel approach was developed for the sensitive, selective and accurate determination of the triclosan, with submicromolar detection limit. Excellent repeatability, reproducibility and long life-time of the prepared sensor were achieved. All the analytical properties of the method can be based on the fabricated homemade composite – improved effective surface area, low particle size and low charge transfer resistance. Based on these facts, successful ap-plication in the real-time sample analysis was shown, with recovery in the range from 99 to 105 %. Thus, the prepared materials can serve as excellent basis for further investigation in this field and potential technology transfer as selected composite act as promising electrocatalyst for the electrochemical determination of triclosan.”

  1. The introduction section should be modified and improved through citing recent references (2021 and 2022) related studies and indicating the novelty of the study compared to the carried works. The following references should be added. Also added citing references regarding TiO2-based composites (Materials Research Bulletin 2018, 98, 314-321; Journal of Solid State Electrochemistry 2018, 22, 129-139; ChemistrySelect 2017, 2, 65-73; International Journal of Environmental Analytical Chemistry 2022, 1-19 (DOI: 10.1080/03067319.2022.2106860).).

- Thank you very much for the suggestion. We improved our Introduction section. Suggested references are provided, as well as, additional ones. 

  1. Check that all references are relevant to the contents of the manuscript.

- Thank you for your suggestion. We carefully checked all the references. Additional references are provided in the revised version, which further improved novelty of our work.

Round 2

Reviewer 1 Report

I think that the revised version of the manuscript might be published.

Reviewer 3 Report

Accept in present form